# Matrix Metalloproteinases Expression Is Associated with SARS-CoV-2-Induced Lung Pathology and Extracellular-Matrix Remodeling in K18-hACE2 Mice

**DOI:** 10.3390/v14081627

**Published:** 2022-07-26

**Authors:** Hila Gutman, Moshe Aftalion, Sharon Melamed, Boaz Politi, Reinat Nevo, Sapir Havusha-Laufer, Hagit Achdout, David Gur, Tomer Israely, Shlomit Dachir, Emanuelle Mamroud, Irit Sagi, Yaron Vagima

**Affiliations:** 1Israel Institute for Biological Research, Ness Ziona P.O. Box 19, Israel; hilag@iibr.gov.il (H.G.); moshea@iibr.gov.il (M.A.); sharonm@iibr.gov.il (S.M.); boazp@iibr.gov.il (B.P.); hagita@iibr.gov.il (H.A.); gurd@iibr.gov.il (D.G.); tomeri@iibr.gov.il (T.I.); shlomitd@iibr.gov.il (S.D.); emmym@iibr.gov.il (E.M.); 2Department of Biological Regulation, Weizmann Institute of Science, Herzel 234, Rehovot P.O. Box 26, Israel; sapir.havusha@weizmann.ac.il; 3Department of Biomolecular Sciences, Weizmann Institute of Science, Herzel 234, Rehovot P.O. Box 26, Israel; reinta.nevo@weizmann.ac.il

**Keywords:** COVID-19, extra-cellular matrix (ECM), K18-hACE2, matrix-metalloproteases (MMP), SARS-CoV-2

## Abstract

The COVID-19 pandemic caused by the SARS-CoV-2 infection induced lung inflammation characterized by cytokine storm and fulminant immune response of both resident and migrated immune cells, accelerating alveolar damage. In this work we identified members of the matrix metalloprotease (MMPs) family associated with lung extra-cellular matrix (ECM) destruction using K18-hACE2-transgenic mice (K18-hACE2) infected intranasally with SARS-CoV-2. Five days post infection, the lungs exhibited overall alveolar damage of epithelial cells and massive leukocytes infiltration. A substantial pulmonary increase in MMP8, MMP9, and MMP14 in the lungs post SARS-CoV-2 infection was associated with degradation of ECM components including collagen, laminin, and proteoglycans. The process of tissue damage and ECM degradation during SARS-CoV-2 lung infection is suggested to be associated with activity of members of the MMPs family, which in turn may be used as a therapeutic intervention.

## 1. Introduction

Nowadays, the coronavirus disease of 2019 (COVID-19) pandemic remains a major concern worldwide with over 250 million cases and over 5 million cumulative deaths according to the World Health Organization (WHO). While the emergence of new variants is of global concern and the campaign towards comprehensive vaccination is at its peak, a global concern regarding the assessment of lung injury for discharged patients remains a major risk factor associated with the development of acute respiratory distress syndrome (ARDS), which can lead to complications such as neutrophilia, organ coagulation, and dysfunction among populations at risk [1]. A significant number of COVID-19 convalescence patients continue to report abnormal pulmonary function seen in lung function tests, lung imaging examination, and exercise capacity [2]. Overall, the mechanisms of lung injury in COVID-19 are associated with cytokine storm and abnormal immune response that leads to pulmonary fibrosis, augmented pro-inflammatory cytokines expression, and tissue damage associated with massive neutrophil infiltration in the lungs [3,4].

The pathophysiology of pulmonary inflammation is characterized by the remodeling of the epithelial–endothelial capillary interface, also known as the blood–air barrier, that consists of alveolar and endothelial cells surrounded by ECM which optimizes the surface area and thickness of the alveolus to provide normal gas exchange [5]. Matrix metalloproteases (MMPs) are a diverse family of extracellular proteinases that participate in many biological processes such as development and tissue homeostasis during health and disease. These zinc-dependent endopeptidases also considered as part of the host-defense arsenal against invading pathogens, implicated in wide variety of infections and capable of degrading ECM components, chemokines, and surface receptors. Nevertheless, their uncontrolled expression due to exacerbation of the immune response in the lungs may lead to severe damage associated with blood–air barrier rupture, ECM disintegration, and pulmonary fibrosis which may lead to organ failure and death in various pulmonary infection diseases [6,7,8,9,10,11].

A fundamental research tool to study the severe acute respiratory syndrome coronavirus 2 (SARS-CoV-2) pandemic are transgenic mice expressing the human angiotensin I-converting enzyme 2 (ACE2) receptor driven by the cytokeratin-18 (K18) gene promoter (K18-hACE2) [12]. These mice develop severe pulmonary disease after intranasal (i.n.) exposure with SARS-CoV-2, accompanied by high levels of pro-inflammatory cytokines and chemokines, leukocytes infiltration into the lung, as well as histopathological pulmonary changes. These changes are associated with injury to the parenchyma, collapse of the alveolar space, and decreased gas exchange which overall resemble severe COVID-19 in human patients [12,13]. The reliability of K18-hACE2 as an animal model of COVID-19 has paved the way to study diverse therapeutic strategies aiming to block SARS-CoV-2 replication, promote its elimination, and enhance host immunization [14,15,16].

Therapeutic studies successfully provided immune protection with assorted vaccination techniques as well as interference with viral replication [16,17,18,19]. However, insights dealing with host–pathogen interactions are lacking. In this study, we observed significant alveolar damage at 5 days post infection (dpi) with a lethal dose of SARS-CoV-2. The tissue damage induced by SARS-CoV-2 was associated with elevated expression levels of MMP8, MMP9, and the membrane-bound MMP14 in the infected lungs. Moreover, a reduction in ECM proteins was observed in the histopathology of lung sections of SARS-CoV-2-infected mice.

Overall, our study provides a link between the pulmonary damage after COVID-19 and MMPs expression in the lung, and may lead to research on therapeutic solutions targeting MMPs during the early and long-lasting effects of SARS-CoV-2 infection.

## 2. Materials and Methods

### 2.1. Cell Lines and Viruses

African green monkey kidney clone E6 cells (Vero E6, ATCC^®^ CRL-1586™) were grown in Dulbecco’s modified Eagle’s medium (DMEM) containing 10% Fetal bovine serum (FBS), MEM non-essential amino acids (NEAA), 2 mM L-Glutamine, 100 Units/mL Penicillin, 0.1 mg/mL Streptomycin, 12.5 Units/mL Nystatin (P/S/N) (Biological Industries, Israel). Cells were cultured at 37 °C, 5% CO_2_ at 95% air atmosphere. For K18-hACE2 transgenic mice infection, we used SARS-CoV-2, isolated Human 2019-nCoV ex China strain BavPat1/2020 that was kindly provided by Prof. Dr. Christian Drosten (Charité, Berlin, Germany) through the European Virus Archive–Global (EVAg Ref-SKU: 026V-03883). Virus stocks were propagated and tittered on Vero E6 cells. The viruses were stored at −80 °C until use. Handling and working with SARS-CoV-2 were conducted in a BSL3 facility in accordance with the biosafety guidelines of the Israel Institute for Biological Research (IIBR).

### 2.2. Animal Experiments

All animal experiments involving SARS-CoV-2 were conducted in a BSL3 facility. Treatment of animals was in accordance with the Animal Welfare Act and the conditions specified in the Guide for Care and Use of Laboratory Animals (National Institute of Health, 2011). Animal studies were approved by the local IIBR ethical committee for animal experiments (protocol numbers HM-07-20 and M-62-20). Female K18-hACE2 transgenic mice (18–20 g, Jackson, Glendora, CA, USA) 8–10 weeks old, and female and male golden Syrian hamsters (60–90 g, Charles River Laboratories, Wilmington, MA, USA) 6–7 weeks old were maintained at 20−22 °C and a relative humidity of 50 ± 10% on a 12 h light/dark cycle. Animals were fed with commercial rodent chow (Altromin, Lage, Germany) and provided with tap water ad libitum. Mice and hamsters were randomly assigned to experiment and kept in groups of 5 and 4, respectively. Infection was performed as previously described [17]. Briefly, SARS-CoV-2 was diluted in PBS supplemented with 2% FBS (Biological Industries, Beit Ha Emek Israel). Animals were anesthetized with a mixture of 0.5% ketamine HCl and 0.1% xylazine and was SARS-CoV-2 suspension was used to infect mice (2000 pfu) and hamsters (5 × 10^6^ pfu) by 20 µL for mice and 50 µL for hamsters, by intranasal (i.n.) instillation of viral suspension.

### 2.3. RT-PCR and Quantitative PCR Analysis

Total RNA from lung cell suspensions was extracted using TRI Reagent (Cat. T9424, Sigma, St. Louis, MI, USA) according to the manufacturer’s instructions. Two micrograms of total RNA were reverse-transcribed using Moloney murine leukemia virus reverse transcriptase and oligo-dT primers (Promega, Madison, WI, USA). Quantitative PCR analysis was performed using an ABI 7500 machine (Applied Biosystems, Waltham, MA, USA) with SYBR Green PCR Master Mix (Applied Biosystems). The fold change in gene transcript quantity compared with hypoxanthine phosphoribosyl transferase (HPRT) was measured using the comparative (–2^∆∆Ct^) method. Forty cycles of PCR were performed in duplicate for each primer from the sequences of primers used in this study: (Table 1).

### 2.4. Collagen Concentration in the BALF-Sircol Assay

The Sircol collagen assay (Biocolor, Carrickfergus, UK) was performed following the manufacturer’s instructions. Briefly, bronchoalveolar lavage fluid (BALF) was collected by exposing the trachea and injecting, then removing, a total of 1 mL PBS containing 1% protease inhibitor cocktail (Sigma) twice. The BALF samples were then filtered and stored at −80 °C. Before analysis, samples were centrifuged at 13,000× *g* for 5 min. One mL of Sirius red reagent was added to each BALF (50 µL) sample and mixed for 30 min. The collagen-dye complex was precipitated by centrifugation at 10,000 rpm for 10 min, and the pellet was dissolved in the supplied alkaline reagent. Finally, the absorbance of the samples was measured at 540 nm. The collagen concentration of the SARS-CoV-2-infected samples expressed as µg/mL was normalized to that of the littermate naive samples.

### 2.5. H&E Staining and Tissue/Air-Space Ratio Calculation

For hematoxylin and eosin (H&E) general histopathology evaluation, lungs were rapidly isolated, and fixed in 4% neutral buffered PFA at room temperature (RT) for 1 week, followed by routine processing for paraffin embedding. Five µm-thick sections were serially cut and selected sections were stained with H&E for light microscopy examination. Images were acquired using 3D HISTECH panoramic midiII3.0.0. Tissue/air space ratio was determined using ImageJ free software analysis (particle analysis algorithm). Images of at least five random regions of interest (ROIs) per section were taken at the same magnification (×20). Color threshold parameters were determined and remained consistent throughout analysis. Total area values were measured separately for air space and tissue. Ratio of total tissue area to total air space area was calculated for each ROI. Average value of at least five ROIs per animal is presented.

### 2.6. Immunofluorescence Staining

Mice and hamsters were euthanized, and the lungs were instilled with 0.8 and 2 mL (respectively) 4% neutral-buffered paraformaldehyde to avoid collapse of the alveoli. Lungs were then rapidly removed, postfixed in 4% neutral-buffered paraformaldehyde (pH = 7.0) at room temperature for 1 week followed by routine processing for paraffin embedding. Then, 5-μm-thick sections were serially cut and selected sections were stained with the following primary antibodies: polyclonal rabbit anti-collagen IV (1:100 Abcam, Cambridge, UK), polyclonal rabbit anti-Laminin (1:100 Thermo Fischer Scientific, Cheshire, UK), Wisteria floribunda lectin (WFA) (1:100 Vectorlabs, Burlingame, CA, USA), monoclonal hamster anti-Podoplanin (T1α) (1:100 Thermo Fischer Scientific, Rochester, IL, USA), polyclonal rabbit anti-Pro-SPC (1:200 Mercury, Burlington, CA, USA), polyclonal rabbit anti-MMP9 (1:200 Abcam, Cambridge, UK), polyclonal rabbit anti-MMP8 (1:100 Proteintech, Chicago, IL, USA), monoclonal rabbit anti MMP14 (1:100 Abcam, Cambridge, UK), monoclonal rabbit anti-CD11b (1:100 Abcam, Cambridge, UK), monoclonal rat anti F4/80 (1:100 Abcam, Cambridge, UK). For SARS-CoV-2 staining we used a primary antibody (in house preparation) of rabbit polyclonal anti-RBD diluted 1:500 with secondary anti rabbit Alexa Fluor 488. For visualization of denatured collagen in hamster lungs we used hybridizing peptide (red-CHP) (20 µM 3Helix, Salt Lake City, UT, USA) and for hamsters MMP14 immunostaining we used LEM2/15 from Irit Sagi’s lab with secondary anti mouse Alexa Fluor 594. Sections were deparaffinized and rehydrated through 100% ethanol, 95% ethanol, 70% ethanol, and 30% ethanol, washed in distilled water and antigens were retrieved using commercial antigen retrieval solution (Dako, Santa Clara, CA, USA). Sections were then permeabilized for 15 min (0.2% Triton X-100 in PBS), blocked for 1 h (10% normal goat serum in PBS containing 0.05% Triton X-100), incubated with primary antibody diluted in antibody dilution buffer (50% blocking solution, 0.05% Triton X-100 in PBS) for 24 h at 4 °C. Sections were then washed three times with washing buffer (1% blocking solution in PBS containing 0.05% Triton X-100) and incubated with anti-rabbit/hamster Alexa Fluor 488or 594 (Molecular Probes, Burlington, CA, USA) for CD11b, MMP8 in mice, Pro-SPC, T1a, collagen IV and laminin, anti rabbit Alexa Fluor 647 (Abcam, Cambridge, UK) for MMP9, MMP14 MMP8 in hamster or Cy3 donkey anti rat (Jackson Immune Research, West Grove, PA, USA) for F4/80. Secondary antibodies were dilute in antibody dilution buffer for 1 h at RT. Nuclei were stained with DAPI. Following three additional washes, slides were mounted using Fluoromount-G (Southern Biotech, Birmingham, AL, USA).

Collagen IV, Laminin, WFA, CHP, T1α, Pro-SPC, CD11b and MMP8 images were acquired and analyzed using a Zeiss LSM710 confocal microscope (Zeiss, Oberkochen, Germany). Murine MMP9, MMP14 and F4/80 images were obtained using spinning disk Nikon CSU-W1 microscope (Nikon Instruments Inc., New York, NY, USA). Set of images were trained and segmented using lastik (1.3.3b2) [20] and Pixel Classifier to classify pixels into Collagen IV, laminin and WFA. The areas of these masks relative to nuclei area (using Otsu threshold) were analyzed using Fiji software [21]. Image analysis for MMP9, MMP14, was performed using Amira software. Nuclei were demarcated by segmenting the DAPI channel using the “Adaptive thresholding”, “Remove small spots”, and “Separate objects” modules. The segmented nuclei mask was segmented and labeled. Each of the labeled nuclei masks was used for individual intensity analysis of the remaining channels of the image.

### 2.7. Statistics

Data were analyzed with GraphPad Prism 7 software. Exact *p* values are provided for each analysis. Statistical significance for weight differences between groups was determined by two-ways ANOVA (group × time, with repeated measure on the latter) followed by an appropriate post hoc analysis (Dunnett). Tissue/Air space ratio analysis statistics was determined by one tailed Mann Whitney nonparametric statistics. For all other analysis significance was determined using Kruskal–Wallis nonparametric test.

## 3. Results

### 3.1. Evaluation of Lung Infection with SARS-CoV-2 in K18-hACE2 Mouse Model

Virulence attributes of SARS-CoV-2 were first evaluated by intranasal (i.n.) infection of K18-hACE2-transgenic mice with a lethal dose of 2000 pfu of SARS-CoV-2. Reduction in total body weight of the infected mice was first observed at 5 dpi (Figure 1A) followed by 75% mortality on day 6, with no survivors by the next day (Figure 1B). Immunostaining of the infected lungs with anti-RBD rabbit polyclonal antibodies indicate the presence of SARS-CoV-2 distribution in the lungs of infected mice at 2 and 5 dpi (Figure 1C). Histopathological changes in the lungs of SARS-CoV-2-infected mice (5 dpi), analyzed by H&E staining of lung sections, exhibited massive leukocytes infiltration (Figure 1(D6), black arrow) and hyperplasia of the bronchial epithelium associated with bronchial congestion, edema, and cellular debris-rich exudates (Figure 1(D7,8) respectively, black arrows). Tissue to air-space ratio analysis in the lungs of SARS-CoV-2-infected mice was higher at 5 days post infection compared to mock, indicative of lung obstruction (Figure 1E). Further analysis of immune cell response as depicted by the staining of the marker CD11b, which expressed on the surface of monocytes, granulocytes, macrophages, and other cells of the innate immune system, revealed massive infiltration of CD11b+ cells in the lungs of SARS-CoV-2-infected mice at 5 days post infection (Figure 1F,G). Altogether, the lung pathology of K18-hACE2 mice subjected to SARS-CoV-2 infection is in line with previously published data and provides a solid platform to further investigate pathological mechanisms involved with SARS-CoV-2 infection.

### 3.2. Characterization of Alveoli Epithelial Damage following SARS-CoV-2 Infection

To further analyze in detail the damage of the alveoli after SARS-CoV-2 infection, we characterized and stained the alveolar epithelium cells. The alveoli are mainly composed of alveolar type I (ATI) cells that cover 95–98% of the alveolar surface, thereby playing a critical role in barrier function and gas exchange, and cuboidal alveolar type II (ATII) cells that induce re-epithelialization of the alveoli during lung injury [22]. ATII epithelial cells in alveoli are among the cell types that co-express the receptor for SARS-CoV-2 cell entry, angiotensin-converting enzyme 2 (ACE2), and its co-receptor transmembrane protease serine 2 (TMPRSS2) [23,24]. We therefore quantified ATII cells and their adjacent ATI cells in the lungs by fluorescent staining of the ATI cell marker T1α and the prosurfactant protein C (proSPC), a marker for ATII cells [25]. As depicted in Figure 2, reduced numbers of ATII cells were observed in the lungs of SARS-CoV-2-infected K18-hACE2 mice by 5 dpi, as indicated by a reduction in proSPC positive cells (Figure 2A,B). Moreover, the levels of ATI cells were also reduced at 5 dpi as revealed by T1α staining pointing on the structural damage to the alveoli (Figure 2A,C).

### 3.3. MMPs Expression Levels in the Lungs of SARS-CoV-2-Infected K-18 Mice

MMPs are major key players during pulmonary diseases associated with either viral or bacterial infections and are considered critical effectors of the host immunity as proteolysis of chemokine, cytokines, adhesion molecules, and ECM components shape the host immune response to infection [8,25,26,27,28]. To evaluate their contribution to the tissue damage after SARS-CoV-2 infection, we quantified the expression levels of different MMPs that were previously documented as highly expressed and active during pulmonary infectious diseases [29]. Quantitative PCR analysis for MMP2, MMP3, MMP7, MMP8, MMP9, MMP12, and MMP14 (MT1-MMP) expression in whole lung extracts obtained from lungs of K18-hACE2 mice at 2 and 5 days post SARS-CoV-2 infection was performed and substantial elevation in mRNA expression of several MMPs was observed (Figure 3A). Further immunostaining of MMP8, MMP9, and MMP14 that exhibited considerable elevation in their mRNA expression was further supported in a protein level as observed by increased expression levels (Figure 3B,C) as well as the total number of MMPs-positive cells (Figure 3D) in the lungs of mice during 5 days post SARS-CoV-2 infection.

The ECM is responsible for tissue organization and function, based on interconnection of matrix molecules that interact with each other to create a dynamic three-dimensional (3D) scaffold for cells to reside within a tissue [30]. In the lungs, the ECM provides a structural platform with contractility properties for epithelial and endothelial cells, enabling the mechanical dynamics of the blood–air interface with normal gas exchange. MMPs have a central role in destructive pulmonary diseases where excessive proteolytic activity causes aberrant degradation of the lung ECM. Although MMPs play important roles in normal pulmonary immunity, in excess they can contribute to immunopathology as a result of ECM components’ degradation and cleavage [26].

### 3.4. Pulmonary ECM Integrity after SARS-CoV-2 Infection

We assessed the expression levels of some major ECM proteins in the lungs of mice infected with SARS-CoV-2. The lungs were extracted from infected mice at the indicated days after infection as well as from mock treated mice, and were subjected to immunofluorescent staining of different ECM components. Five days after infection, a significant reduction in the manifestation of collagen type IV (Col.IV), laminin, and proteoglycans (WFA) were observed in the lungs of K18-hACE2 mice exposed to SARS-CoV-2 (Figure 4A–C,E–G). Masson’s trichrome staining of lung sections demonstrated the accumulation of fibrillar collagen fibers in the bronchi of infected mice at 2 and 5 days post infection (Figure 4D). In accordance, increased levels of free collagen fibers detected in the BALF (broncho-alveolar lavage fluid) of SARS-CoV-2-infected mice at 2 and 5 days post SARS-CoV-2 infection (Figure 4H,I). These findings further supported the decreased levels of intact Col.IV in lung sections of infected mice as demonstrated by the accumulation of collagen degraded fibers that were found in the BALFs of the mice.

## 4. Discussion

The findings of this study point out the involvement of the MMPs family members and the pulmonary pathology of SARS-CoV-2-infected K18-hACE2 mice. In this study, the disease progression of SARS-CoV-2 infection in K18-hACE2 transgenic mice is consistent with other studies using hACE2 mice. However, although the mice exhibited a similar degree of weight loss and poor survival rates, it should be noted that the varying degree of symptoms as well as histopathological damage may be explained by differential host immune response as well as the resident microbiota between individual mice as previously suggested [31]. Pulmonary viral infection with a dose of 2000 pfu resulted in weight loss 5 days post infection and mortality of all infected animals by day 7 post infection. In terms of lung tissue damage, we identified hyperplasia of the bronchial epithelium including bronchial congestion, edema, and protein-rich exudates which prompted a high tissue to air-space ratio in SARS-CoV-2-infected mice compared to mock. In line with the above findings, we found an increase in immune cells infiltration as depicted by CD11b staining, which is regarded as a marker for monocyte/macrophages and granulocytes, and a reduction in ATII cells in the lungs after SARS-CoV-2 infection were also observed. These pathological findings are in accordance with postmortem findings in patients who died from COVID-19 pneumonia that exhibited injury to the alveolar epithelial cells, hyperplasia of type II pneumocytes, fibrin forming clusters in airspaces, and abundant intra-alveolar neutrophilic infiltration [32]. Based on these histopathology characteristics, we decided to further analyse the involvement of members of the MMPs family during SARS-CoV-2 infection, given that their deleterious effect to the lung during viral or bacterial infection was previously demonstrated [6,25,27,33,34]. Quantitative PCR and immunofluorescent staining of infected lungs revealed increased expression of MMP8, MMP9, and MMP14 associated with macrophages (F4/80 positive cells) in the lungs of SARS-CoV-2-infected K18-hACE2 mice (Appendix A). The connection between the immune cells’ infiltration into the lungs and MMP8, MMP9, and MMP14 over expression after SARS-CoV-2 infection is in line with cumulative studies that emphasize the involvement of MMPs expression by immune cells. MMPs secretion by both recruited and resident immune cells is considered as part of innate and adaptive immune function during pulmonary infectious diseases [5,7].

To evaluate the tissue damage associated with the elevation in MMPs expression in the lungs, we characterised the expression of major ECM components that are considered as MMPs substrates. Based on immunofluorescent staining of the infected lungs we observed reduction in ECM elements in the lungs, including collagen IV, laminin, and proteoglycans as depicted by WFA staining. Accordingly, an increase in cellular debris and collagen fibres around the bronchi as depicted by Masson’s trichrome staining and accumulation of free collagen fragments in the BALF were also detected. These observations were further corroborated in another animal model for COVID-19 in humans, namely, Syrian golden hamsters [35,36,37]. Disease progression in this non-lethal model was characterized by severe lung damage, accompanied by increased MMP8 and MMP14 levels, followed by an increase in ECM remodelling in the lungs (Appendix A).

Robust immune response in the lungs, as reflected by excessive leukocytes infiltration and proteolytic activity, impaired tissue homeostasis and ECM integrity, eventually leading to pathological outcome such as ARDS. Host-derived inflammatory response followed by proteases expression provides a delicate balance between the beneficial and detrimental effect on organ integrity and functionality, while overexpression may result in severe tissue damage and organ failure [29]. In this study we focused on MMPs expression in the lungs after SARS-CoV-2 infection. Interestingly, recent publications shed new light on the prognostic role of several MMPs accumulation in the plasma of COVID-19 patients. High serum levels of MMP7 and MMP9 and the levels of macrophages activation factors, and severity of COVID-19 in obese-diabetic patients were associated with ARDS in COVID-19 patients with risk factors [38].

In another study, the comparison between the levels of inflammatory factors among hospitalized COVID-19 patients exhibited a positive correlation between serum levels of VEGFA and MMP1, suggesting that MMP1 has a critical role in tissue remodeling and vascular permeability. These findings imply an interaction between serum MMPs and endothelial cells in COVID-19 patients, and an association with the severity of the disease [39]. As such, the plasmatic levels of MMP2 and MMP9 in patients with severe COVID-19 documented a significant association between their levels and disease severity with the potential to predict the risk of in-hospital death [40]. Taken together, these cumulative pieces of evidence that link the circulating immune cells and serum levels of MMPs in COVID-19 patients are supported by a study that showed that lymphopenia in severe COVID-19 patients attributed to immune cell migration into inflamed tissues [41]. In accordance, here we show that SARS-CoV-2-infected lungs exhibited massive infiltration of immune cells from peripheral blood, which are suggested to be the source of MMPs in the lungs that mediate the degradation of ECM components and may enable the collapse of the blood–air barrier in the lungs.

ECM integrity in the lungs provides structural support and contractility for the cells residing the lower airways compartment. A proteomic analysis of clinical samples obtained from COVID-19 patients indicated that major ECM components, including heparan sulfate proteoglycans, collagen VI, laminins, annexin A2, and fibronectin, were diminished in the COVID-19 lungs and may thus be the reason for the impaired elasticity of the bronchial wall, bronchial epithelial cell attachment, and overall alveolus functionality [42]. Collagen, laminin, and other ECM components are considered as MMPs substrates and described as targets of various MMPs during different chronic and acute lung diseases [7]. Our findings support the assumption that increased release of MMPs may accelerate the damage observed in the lungs of COVID-19 patients and suggest that MMPs proteolytic activity during and after SARS-CoV-2 infection may become a potential target for COVID-19.

## Figures and Tables

**Figure 1 viruses-14-01627-f001:**
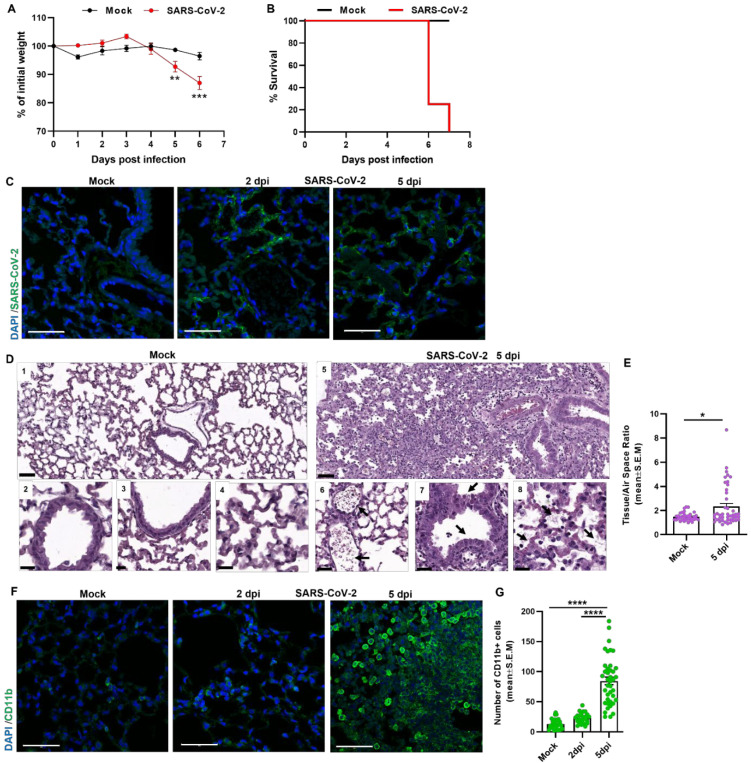
K18-hACE2 mouse model for airways infection with SARS-CoV-2. (**A**) Body weight changes of i.n mice infected with 2000 pfu of SARS-CoV-2 compared to mock (n = 6 for Mock and SARS-CoV-2, the error bars represent the SEM), and (**B**) survival rate as depicted by Kaplan–Meier survival curves (n = 20). Weight differences between groups were analyzed by two ways ANOVA (group × time, with repeated measure on the latter) followed by an appropriate post hoc analysis (Dunnett) using GraphPad 7. ** *p* < 0.01, *** *p* < 0.001. (**C**) Representative SARS-CoV-2 immunolabeling of lung sections of mock and SARS-CoV-2-infected mice at 2 and 5 dpi. (Scale bar 50 µm). (**D**) H&E staining of lung sections from K18-hACE2 mice, mock or 5 days post infection (dpi). Upper images show low magnification (scale bar 100 µm) and lower images show high magnification (scale bar 50 µm). (**E**) Tissue/Air space ratio analysis of mock and SARS-CoV-2-infected mice (5 dpi), (n = 7 for Mock and n = 8 for SARS-CoV-2), for each mouse five fields were imaged and analyzed. The error bars represent the SEM. Statistical significance was calculated using one tailed Mann Whitney non-parametric statistics * *p* < 0.05. (**F**) Representative immunofluorescence images of 5 µm lung sections labeled for immune cells marker-CD11b (green) of mock and SARS-CoV-2-infected mice at 2 and 5 dpi. (Scale bar 50 µm). (**G**) Quantitative analysis of CD11b+ cells in lung sections of mock and SARS-CoV-2-infected mice at 2 and 5 dpi (n = 6 for mock and n = 5 for 2 and n = 8 for 5 dpi), for each mouse five fields were imaged and analyzed. The error bars represent the SEM. Statistical significance was Kruskal–Wallis nonparametric test. **** *p* < 0.0001.

**Figure 2 viruses-14-01627-f002:**
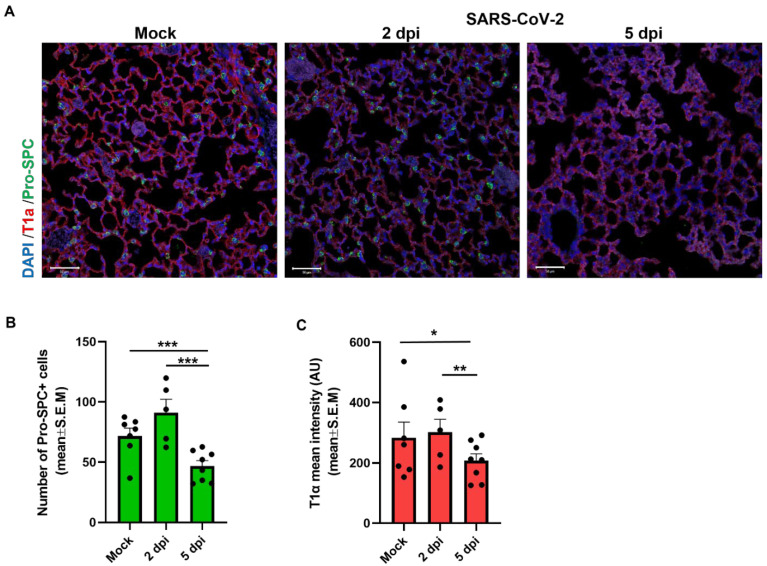
Immunophenotype of alveolar epithelial cells in lung sections of SARS-CoV-2-infected K18-hACE2 mice. K18-hACE2 mice were infected with 2000 pfu of SARS-CoV-2 compared to mock. (**A**) Representative immunofluorescence images of lung sections labeled for alveolar type I (red) and type II (green) cells at 2 and 5 days post SARS-CoV-2 infection compared to mock treated mice. (**B**) Quantitative analysis of number of type II epithelial cells (Pro-SPC+). (**C**) Mean fluorescence intensity of type I epithelial (T1α+) cells expression in lung sections of mock and SARS-CoV-2-infected mice at 2 and 5 dpi (n = 7 mice for mock and n = 5 for 2 dpi and n = 8 for 5 dpi). Scale bars: 50 µm. Data presented with analysis of five fields per mouse. Statistical significance was calculated using Kruskal–Wallis nonparametric test. * *p* < 0.05, ** *p* < 0.005, *** *p* < 0.0001.

**Figure 3 viruses-14-01627-f003:**
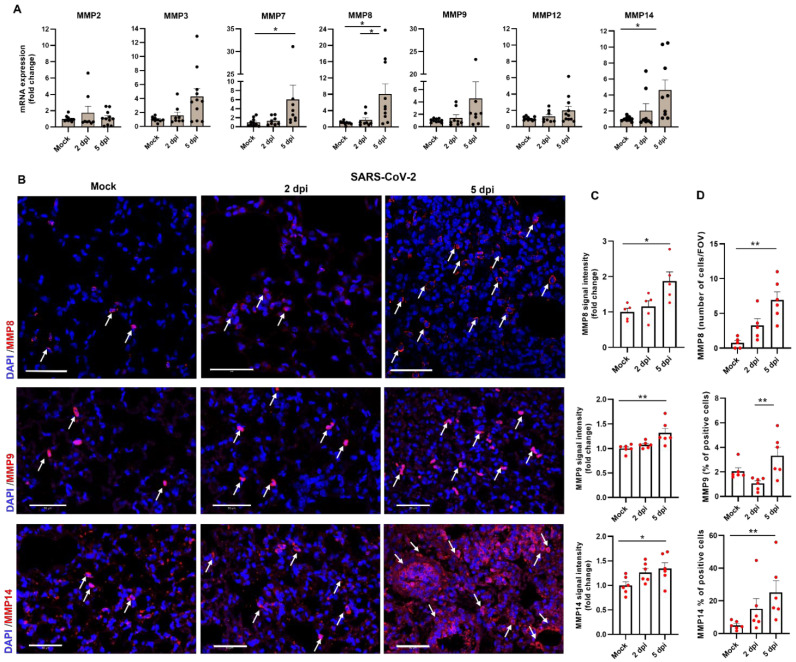
Increased MMPs expression in the lungs of K18-hACE2 mice following SARS-CoV-2 infection. (**A**) Quantitative RT-PCR analysis of MMP mRNA in the lungs of K18-hACE2 mice 2 and 5 dpi compared to mock. Data presented as mean ± SEM (n = 8–11 mice per time point). (**B**) Representative immunofluorescence images of lung sections labeled for MMP8 (upper panel), MMP9 (middle panel) and MMP14 (lower panel) at 2 and 5 days post SARS-CoV-2 infection compared to mock Scale bar: 50 µm. Arrowheads indicate MMPs labeled cells. (**C**) Mean fluorescence signal intensity of MMP8, MMP9, and MMP14 and (**D**) number of cells for MMP8 and percent of positive cells for MMP9 and MMP14 in lung sections of mock and SARS-CoV-2-infected mice at 2 and 5 dpi (n = 5–6 mice for mock and n = 5–6 for 2 dpi and n = 5–6 for 5 dpi. For MMP8 analysis, 5 fields were imaged for each mouse for MMP9 and MMP14 analysis 25 fields were imaged for each mouse. Statistical significance was calculated using Kruskal–Wallis nonparametric test * *p* < 0.05, ** *p* < 0.01.

**Figure 4 viruses-14-01627-f004:**
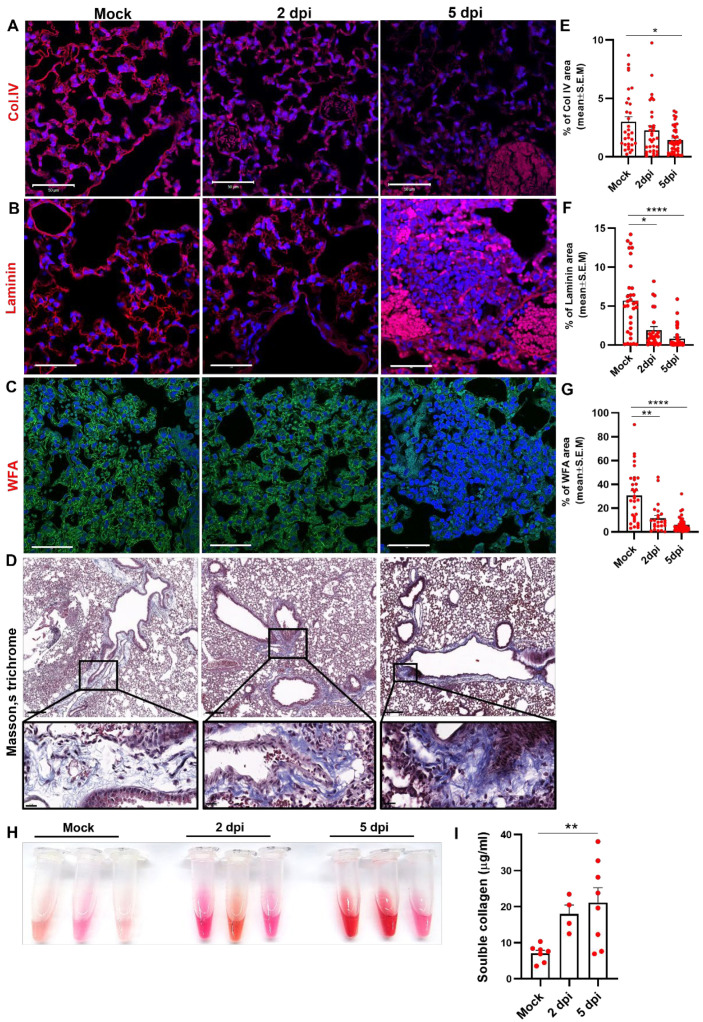
SARS-CoV-2 infection induce lung ECM remodeling in K18-hACE-2 mice. Representative images and the corresponding quantification of lung sections labeled for collagen IV (**A**,**E**), laminin (**B**,**F**), WFA (**C**,**G**) and Masson’s trichrome (**D**) (MTC) at 2 and 5 days post SARS-CoV-2 infection compared to mock. For each group n = 6–8 mice and for each mouse, five fields were imaged and analyzed. Scale bar: (**A**–**C**) 50 µm, (**D**) 200 µm, inset 20 µm. (**H**) Representative soluble collagen accumulation in BALF samples of mice 2 and 5 days post SARS-CoV-2 infection compared to mock. (**I**) Quantification of soluble collagen concentrations in the BALF samples; (n = 6 mice for mock and n = 5 for 2 dpi and n = 8 for 5 dpi). The error bars represent the SEM. Statistical significance was calculated using Kruskal–Wallis nonparametric test * *p* < 0.05, ** *p* < 0.005, **** *p* < 0.0001.

**Table 1 viruses-14-01627-t001:** Sequences of the primers used in this study.

Mouse Gene	Forward 5′-3′	Reverse 5′-3′
MMP2 NM_008610	ACCATGCGGAAGCCAAGAT	TTAAGGCCCGAGCAAAAGC
MMP3 NM_010809	CTTCCCAGGTTCGCCAAAAT	CATGTTCTCCAACTGCAAAGGA
MMP7 NM_010810	GGTGAGGACGCAGGAGTGAA	GCGTGTTCCTCTTTCCATATAACTTC
MMP8 NM_008611	CACACACAGCTTGCCAATGC	TCCCAGTCTCTGCTAAGCTGAAG
MMP9 NM_013599	CAGACGTGGGTCGATTCC	TCATCGATCATGTCTCGC
MMP12 NM_008605	TGAGGCAGAAACGTGGACTAAA	GGGCTCCATAGAGGGACTGAA
MMP14 NM_008608	CCCAAAAACCCCGCCTAT	TCTGTGTCCATCCACTGGTAAAA
HPRT-1 NM_013556	AGTACAGCCCCAAAATGG	TCCTTTTCACCAGCAAGCT

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
