# Peer review of "Matrix Metalloproteinases Expression Is Associated with SARS-CoV-2-Induced Lung Pathology and Extracellular-Matrix Remodeling in K18-hACE2 Mice"

_viruses, 2022, doi:10.3390/v14081627_

Round 1

Reviewer 1 Report

The MS by Gutman et al evaluate the modulation of MMP expression in the lung of hACE2 transgenic mice infected by lethal doses of SARS-CoV-2. Authors also establish correlation between increase of selected MMPs with tissue damage and ECM protein degradation.

Experiments are well performed, presentation clear and discussion interesting, but the study would require more results to validate the title.

Major concern

1) The data provided as very limited and descriptive. Correlation mentioned is not statistically validated. Authors should analyse at in a multiparametric analysis to confirm their conclusion. A better characterization of immune cell infiltration should be included as CD11b staining is not really precise, as mentioned by the authors themselves.

2) The model used, hACE2 K18 mouse, is know known to develop extremely rapidly SARS-CoV-2 induced severe damages, that does not mimic the situation in human, and in addition authors use high doses of viruses for IN inoculation. Therefore the outcome of this infection is of limited interest for human COVID.

However, authors performed in parallel a second study on golden Syrian hamster, from which only MMP expression is shown in Fig S1. This model with slower development of COVID is more appropriate for understanding for human COVID.

Thus authors should complete this study by comparing similar data that can obtain with K18-mice to that with Syrian hamster and establish a potential correlation between variation of the parameter studied and the severity of the infection.

Other concerns:

1) The title does not reflect the findings reported that are much more trivial as well reported in the abstract. Title should be changed to be more factual unless more data are added. One would like that authors demonstrated the message of the title though, but therefore, functional analyses should have been performed.

2) Figure 1E,G: it seems that the sections from the mouse group infected for 5days splits into two subgroups.   Do the highest points shown belong to a subset of mice? Or reflects heterogeneity of the tissue? Especially in E, where the differences are due to around 25% of the section analysed, whereas the 75% remains identical to the mock group.

3) line 227: Authors mention ciliated  cells but them refer to those cells as ATI when analysing marker T1a. Please clarify.

4) Figure 2: ATI cells represents >95% of cells from the alveoli (also stated in line 225) but the number of cell counted per section in B and C is more 20% of Pro-SPC+ ATII cells (value in the range of 70) and 80 of T1a+ ATI (value around 260). It may be that these values do not represent cells but intensity labelling: then how authors can compare the two markers?

Due to the heterogeneity of the mock mouse group, it is very difficult to see an effect.  Do the outlier points correspond to the same animals in the two graphs B and C? Did these animals develop specific signs unrelated to COVID?

5) Quantification of Figure 3B us missing.

Author Response

Response to reviewer 1

Comment 1: The data provided as very limited and descriptive. Correlation mentioned is not statistically validated. Authors should analyse at in a multiparametric analysis to confirm their conclusion.

We did not intend to suggest statistical correlations between the tissue damage and the MMPs activity. We thus corrected the relevant section in the text. As requested by the reviewer, a multiparametric analysis was performed (see material and methods section of the revised manuscript and figure legends in accordance). The additional statistical analysis confirmed the conclusions described in the original article.

A better characterization of immune cell infiltration should be included as CD11b staining is not really precise, as mentioned by the authors themselves.

As requested, we preformed additional staining of immune cells in the lung after SARS-CoV-2 infection. We now demonstrate co-localization of MMP14 with F4/80 (Macrophages) positive cells. The new data was added to the revised manuscript (Sup. Fig 1). With regard to MMP8 and MMP9, these MMPs are secreted by both neutrophils and macrophages which share the leukocyte-specific receptor – CD11b, which is regarded as a marker for monocyte/macrophages and granulocytes (Please see discussion - line 374).

Comment 2: The model used, hACE2 K18 mouse, is now known to develop extremely rapidly SARS-CoV-2 induced severe damages, that does not mimic the situation in human, and in addition authors use high doses of viruses for IN inoculation. Therefore, the outcome of this infection is of limited interest for human COVID. Thus, authors should complete this study by comparing similar data that can obtain with K18-mice to that with Syrian hamster and establish a potential correlation between variation of the parameter studied and the severity of the infection.

The hACE2-K18 mouse model is a powerful research tool adopted by many researchers worldwide to study of the pathogenesis of COVID-19 and to develop countermeasure approaches to combat SARS-CoV-2 propagation among humankind (Fatai S. Oladunni, Nat. Com. 2020, Winkler ES et al., Nat. Immu. 2020, Joyce MG et al., Cell Reports 2021, chen RE et al., Nature 2021). We agree with the reviewer and aware to the fact that this animal model has its own limitations. However, the hACE2-K18 model described in this work, as well as in many other publications, enabled to obtain reproducible results with 2000 pfu of viral inoculation and allowed us to shed a new light on the mechanism of lung tissue damage induced by matrix-metalloproteases after pulmonary SARS-CoV-2 infection. On the other hand, Golden Syrian hamster develop very mild diseases symptoms although infected with a very high dose of 5x106 pfu, an infection dose which is 2500 times higher than the dose administered to the hACE2-K18 mice. Hence, we find the hACE2-K18 mouse model suitable although its limitations to recapitulates histopathological findings of COVID-19 associated ARDS.

Comment 3: The title does not reflect the findings reported that are much more trivial as well reported in the abstract. Title should be changed to be more factual unless more data are added. One would like that authors demonstrated the message of the title though, but therefore, functional analyses should have been performed.

We have changed the title according to the reviewer suggestion. The new title is: “Matrix metalloproteinases expression is associated with SARS-CoV-2 induced pulmonary extracellular-matrix remodeling in K18-hACE2 mice”.

Comment 4: Figure 1E,G: it seems that the sections from the mouse group infected for 5days splits into two subgroups.   Do the highest points shown belong to a subset of mice? Or reflects heterogeneity of the tissue? Especially in E, where the differences are due to around 25% of the section analysed, whereas the 75% remains identical to the mock group.

The tissue damage observed in this model and the hamster model as described elsewhere (Yahalom-RonenY et al 2020) is characterized by focal lesions right next to areas of intact tissue which generate heterogeneity of the lung pathology. This is demonstrated as two subgroups within the infected group not seen in the mock group (1E,G).

As suggested, the data presented in Fig. 1 E was re analyzed using one tailed Mann Whitney non parametric statistics.

Comment 5: line 227: Authors mention ciliated cells but them refer to those cells as ATI when analysing marker T1a. Please clarify.

We thank the reviewer for his comment. This issue is now clarified in the text and the term “Ciliated cells” has been removed to avoid confusion, as we focus mainly at lower respiratory tract and alveoli in particular (Lines 256-261)

Comment 6: Figure 2: ATI cells represents >95% of cells from the alveoli (also stated in line 225) but the number of cell counted per section in B and C is more 20% of Pro-SPC+ ATII cells (value in the range of 70) and 80 of T1a+ ATI (value around 260). It may be that these values do not represent cells but intensity labelling: then how authors can compare the two markers?

In both figures:  2B and 2C,  ATI cells were labeled with anti T1α (Podoplanin) a marker of Alveolar Type I cells and for anti-pro-SPC, a surfactant protein that characterize alveolar ATII cells. Both markers were previously described by other and us (Jansing NL et al 2017, Vagima Y et al., 2020).  This cell staining approach allows to calculate cells proportion and tissue localization per field of view (FOV) to demonstrate the overall change of each cell type in the context of the tissue, compared to non-infected tissues. We didn’t compare ATI cells to ATII cells but rather each cell population to itself along disease progression. The data in both figures represent the number of cells for TypeII and intensity for the marker of Type I epithelial cells, based on the histopathology morphology of the lungs after SARS-CoV-2 infection.

Comment 7: Due to the heterogeneity of the mock mouse group, it is very difficult to see an effect.  Do the outlier points correspond to the same animals in the two graphs B and C? Did these animals develop specific signs unrelated to COVID?

The outlier points in Fig 2 B and C are not from the same animal. All animals infected showed similar disease progression and 80% of the mice died by 6 days post infection and the others died at day 7 (Fig 1B). We cannot attribute any unique signs to these animals.

Comment 8: Quantification of Figure 3B us missing.

Done as requested.

Reviewer 2 Report

To authors

This manuscript was described about the mechanism of lung injury by SARS-CoV-2 infection. Authors focused MMPs and had studied, using animal models with COVID-19. It will be a first report that plural MMPs in COVID-19 were evaluated. I have some questions and comments.

Comments

1.     Materials and Methods

a)     Statistical analysis was not described in the text. Please described detail information of statistical analyses in the text, such as types of statistical analysis and softwares (name and version).

2.     Results

a)     Authors evaluated the differences between groups by ANOVA. When samples show the normal distribution, ANOVA will be used. However, samples in this study will not show the normal distribution, because the number of samples were small, and outliers were included. If samples did not show the normal distribution, non-parametric statistical method should be used, not ANOVA.

b)    Significant higher MMP3, 8 and 14 were shown in Figure 3. Although the relationship between remodeling and MMP14 was evaluated in Supplement figure 1, MMP3 or 8 were not evaluated. Authors should evaluate the role of MMP3 and 8 in ECM remodeling .

3.     Discussion

a)     According to new results analyzed by non-parametric statistical method, authors should re-discuss.

b)    Lower MMP2 and higher MMP-9 in patients with COVID-19 were reported by former publications. Authors citied those publications in Discussion. However, the animal model in this study, MMP3, 8 and 14 showed significant higher, but not MMP-2 and 9. Authors should discuss why MMP3 and 8 were higher in this model, and why not MMP2 and 9.

Author Response

Comment 1: Statistical analysis was not described in the text. Please described detail information of statistical analyses in the text, such as types of statistical analysis and softwares (name and version).

Done as requested. Detailed statistical analysis information is now provided in Material and Methods section.

      Comment 2: Authors evaluated the differences between groups by ANOVA. When samples show the normal distribution, ANOVA will be used. However, samples in this study will not show the normal distribution, because the number of samples were small, and outliers were included. If samples did not show the normal distribution, non-parametric statistical method should be used, not ANOVA.

       As requested by the reviewer, we re-analyzed the data using Kruskal-Wallis nonparametric test, please see material and methods section of the revised manuscript and figure legends in accordance. The additional statistical analysis confirmed the conclusions described in the original article.

Comment 3: Significant higher MMP3, 8 and 14 were shown in Figure 3. Although the relationship between remodeling and MMP14 was evaluated in Supplement figure 1, MMP3 or 8 were not evaluated. Authors should evaluate the role of MMP3 and 8 in ECM remodeling.

       As requested by the reviewer, a new staining of pulmonary MMP8 expression in the lungs of SARS-CoV-2 infected hamsters is now provided (see supplement Figure 2 of the revised manuscript). Regarding MMP3 and MMP9 staining in hamster’s lungs, we encountered reagent limitation as we observed non-specificity staining among different commercial antibodies we’ve been trying. Hence, we cannot provide this data in our manuscript.

       Comment 4: According to new results analyzed by non-parametric statistical method, authors should re-discuss.

A new analysis of the data was performed according to reviewers’ suggestion and the new statistical analyses confirmed the conclusions described in the original article.

Comment 5: Lower MMP2 and higher MMP-9 in patients with COVID-19 were reported by former publications. Authors citied those publications in Discussion. However, the animal model in this study, MMP3, 8 and 14 showed significant higher, but not MMP-2 and 9. Authors should discuss why MMP3 and 8 were higher in this model, and why not MMP2 and 9.

MMP2 and MMP9 detection in the human serum is a clinical tool to diagnose various viral and bacterial infection diseases as well as different cancer associated diseases. Referring to the publications discussed, MMP9 and 2 enzymatic activity was measured in the plasma of COVID-19 patients as a factor that correlates to the disease severity which may be a result of a significant increase in leukocytes (monocytes and neutrophils) in the blood circulation. In our model we present more sever disease which is associated with pulmonary damage and all of the infected animals succumbed by day 7. Further, we tested the MMPs at the site of infection (the lung) and not in plasma or serum peripherally to the site of infection. In this time frame, both mRNA and protein levels of MMP9 in the lung (and not plasma or serum) were increased by 5dpi (Figure 3A-C). Regarding MMP2, we did not observe any significant change in the mRNA levels in the lungs which might be due to the differences in disease progression in mice compared to humans.

Reviewer 3 Report

Title: Matrix metalloproteinases facilitate SARS-CoV-2 induced lung pathology characterized by extracellular-matrix remodeling

Abstract

The abstract presented the study using a straightforward layout. The authors performed a brilliant work showing its objectives, methods, and main findings briefly, without distracting the reader from the important conclusions.

Keywords: the authors can cite the words in alphabetical order

Introduction

i) To revise the abbreviations.

ii) The authors include a figure demonstrating the MMP action and its association with the COVID-19 disease. 

iii) The following excerpt needs to have references: “Therapeutic studies successfully provided immune protection with assorted vaccination techniques as well as interference with viral replication.”

iv) Here, “In this study, we observed a significant alveolar damage, massive inflammatory cells infiltration, bronchial congestion and protein-rich exudates associate with changes in tissue/air space ratio and reduction in alveolar type I (ATI) and type II (ATII) epithelial cells, by 5 days post infection (dpi) with a lethal dose of SARS-CoV-2. This lung pathology was associated with elevated levels of MMP8, MMP9 and the membrane bound MMP14 in the infected lungs. In line, reduction in ECM proteins was observed in histopathology of lungs sections of SARS-CoV-2 infected mice.”, the authors are presenting its results. This excerpt should be moved to results section.

Methods 

ml should be mL

μl should be μL

unpaired student t test should be unpaired student T-test

to delete “virus” after “SARS-CoV-2 virus”

Table 1. The title is missing

I did not have any additional comments on the experimental design. The study was well conducted. Only, in my opinion, could the authors include a figure demonstrating the study design and the analysis performed in each experiment.

Results

The figures have low resolution. Also, the authors should improve the size bar for each image (it is not possible to visualize the scale bar)

Figure 2. To separate figure legends (B) and (C)

Discussion

No comments. The authors discussed the main findings of the study. However, the authors should include the limitations of the study and discuss its impact in the article's main findings.

Author Response

Answer to reviewer 3:

Comment 1: Keywords: the authors can cite the words in alphabetical order

Done as requested

Comment 2: Introduction: To revise the abbreviations.

Done as requested

Comment 3: Introduction: The following excerpt needs to have references: “Therapeutic studies successfully provided immune protection with assorted vaccination techniques as well as interference with viral replication.”

Done as requested

Comment 4: Introduction: Here, “In this study, we observed a significant alveolar damage, massive inflammatory cells infiltration, bronchial congestion and protein-rich exudates associate with changes in tissue/air space ratio and reduction in alveolar type I (ATI) and type II (ATII) epithelial cells, by 5 days post infection (dpi) with a lethal dose of SARS-CoV-2. This lung pathology was associated with elevated levels of MMP8, MMP9 and the membrane bound MMP14 in the infected lungs. In line, reduction in ECM proteins was observed in histopathology of lungs sections of SARS-CoV-2 infected mice.”, the authors are presenting its results. This excerpt should be moved to results section.

Done as requested

Comment 5: ml should be mL, μl should be μL, unpaired student t test should be unpaired student T-test, to delete “virus” after “SARS-CoV-2 virus”, Table 1. The title is missing

Done as requested

Comment 6: The figures have low resolution. Also, the authors should improve the size bar for each image (it is not possible to visualize the scale bar) Figure 2. To separate figure legends (B) and (C)

Done as requested

Comment 7: However, the authors should include the limitations of the study and discuss its impact in the article's main findings.

As requested, the limitations of this study are discussed in the article’s main findings (lines 347-353)

Round 2

Reviewer 2 Report

My all questions were solved. I have no additional questions.